# Quantifying Information Disclosure During Gradient Descent Using Gradient Uniqueness

## Abstract

Disclosing private information via publication of a machine learning model is often a concern. Intuitively, publishing a learned model should be less risky than publishing a dataset. But how much risk is there? In this paper, we present a principled disclosure metric called *gradient uniqueness* that is derived from an upper bound on the amount of information disclosure from publishing a learned model. Gradient uniqueness provides an intuitive way to perform privacy auditing. The mathematical derivation of gradient uniqueness is general, and does not make any assumption on the model architecture, dataset type, or the strategy of an attacker. We examine a simple defense based on monitoring gradient uniqueness, and find that it achieves privacy comparable to classical methods such as DP-SGD, while being substantially better in terms of (utility) testing accuracy.

## 1 Introduction

Data privacy is a crucial concern in data analytics and machine learning. The most commonly-applied definition of privacy is "differential privacy" (DP) (Dwork et al., 2006; Dwork & Roth, 2014). Intuitively, for a randomized algorithm to be differentially private, it must be the case that, whenever the algorithm is run on two input datasets $A$ and $B$ that are similar, the output of the algorithm is likely to be the same. That is, it should not be possible to construct a set of outputs $S$ such that the output of the algorithm on $A$ is much more (or less) likely to be in $S$ than the output of the algorithm on similar dataset $B$.

DP as a definition deals with what happens in the very worst case. It ensures that there exists no way for an attacker to choose a set of possible outputs that can distinguish between *any* two similar input datasets. One problematic implication of the pessimism is that methods to ensure DP can be extremely disruptive. In this paper, we are concerned with mini-batch stochastic gradient descent (SGD), which powers modern AI. The differentially-private version of SGD (DP-SGD) (Abadi et al., 2016) requires adding noise to gradients as well as clipping gradients (limiting their magnitude) so that no datapoint can influence a gradient too much. Unfortunately, this is a very onerous change to the algorithm. Training a large language model (LLM) like Gemini (Google, 2025) or Llama (Grattafiori et al., 2024) can easily cost millions of dollars, and few organizations who will spend such sums will accept adding noise and truncating each and every gradient, out of concern to the quality of the resulting model. As a result, the state-of-the-art differentially private LLM (Vault-Gemma, 2025) has performance on par with GPT2 (Radford et al., 2019), a full model generation before GPT3's capabilities captured widespread public attention (Brown et al., 2020).

Because of the impracticality of DP as a privacy metric in such situations, there has been some recent interest in new definitions of privacy that do not make worst-case assumptions. For example, one may assume a prior over datasets (in the Bayesian sense (Triastcyn & Faltings, 2020)), so that the privacy definition need not be concerned with all possible inputs, only likely ones. Such definitions may allow for less onerous privacy preservation algorithms.

However, having *any* definition of privacy that requires a user to modify SGD to match that definition is a problem if the user is unwilling to accept a change to SGD. Thus, in this paper, we look at the problem in a very different way. Specifically, we ask: Given a *particular* dataset being used to power a mini-batch SGD algorithm, can we rigorously quantify the level of disclosure by the SGD algorithm? Mini-batch SGD has some inherent, privacy-preserving characteristics, even without

modifying the algorithm at all. It is a stochastic algorithm, where randomly selected per-point gradients are summed together, and depending upon the exact scenario, an attacker will typically be able to see only final model. This naturally limits disclosure. If it is determined that points are not safe, they can then be protected by auditing the result of queries to the model (in a closed-source model). For open-source models, methods such as machine unlearning (Bourtoule et al., 2021) can be used. Or, one could simply monitor information disclosure during training and drop those points with highest disclosure. Dropping a few percent of the input data may be more acceptable than changing the training algorithm by adding noise. We explore this option experimentally.

**Our Contributions.** We present a mathematically derived privacy score called *gradient uniqueness* or GNQ, that monotonically increases with an upper bound on the information disclosed to an at-tacker by mini-batch gradient descent. Thus, GNQ provides a principled, well-justified analysis of the risk of the disclosure of individual datapoints during SGD. Specific contributions are:

• The basis of GNQ in a mathematical analysis of SGD contrasts with other, heuristic privacy scores used in empirical attack-based auditing (Sec. 2). The derivation of GNQ is general, and does not make any assumption on the model architecture, dataset type, and the adversary strategy (Sec. A).

• We show empirically the ability of GNQ to explain/predict the success rate of various attacks, as well as the impact of standard, algorithmic components of the SGD algorithm (batch size, learning rate, etc.) on the vulnerability of a learned model. (Sec. 6.2)

• While the derivation of a suite of defenses based on GNQ is left to future work, we empirically test a simple, GNQ-based defense: rank datapoints within a dataset from the highest risk to the lowest risk using GNQ, and remove those at highest risk from training. We provide a compute/memory-efficient implementation suitable for large models and datasets. (Sec. 5)

• Our experiments consider CNN-based models like the ResNet-family and Transformer-based mod-els like the BERT-family. We find that a GNQ-based defense achieves comparable privacy level as DP-SGD; nearly perfect protection from a membership inference attack (MIA), while being sub-stantially better in terms of model utility. It also protects against reconstruction attacks like model inversion while maintaining high model accuracy. (Sec. 6.1)

## 2 RELATED WORK

**Empirical Attack-based Privacy Auditing.** Most auditing approaches probe a trained model with explicit attacks, then interpret attack success as evidence of privacy risk (Murakonda & Shokri, 2020; Nicolae et al., 2018; Trail of Bits, 2021; Kazmi et al., 2024). Common attack types and their decision statistics (a.k.a privacy scores) include: (i) shadow-model MIA (Shokri et al., 2017), which feeds the model's posterior/confidence vector (often top-$k$ probabilities) to an attack classifier; the per-example score is the attack model's membership probability; (ii) white-box MIAs (Nasr et al., 2019), which use gradients, activations, and per-layer norms as features; the score is a classifier over these white-box features; (iii) loss-based MIAs (Sablayrolles et al., 2019; Watson et al., 2022), which take the (negative log-)loss on the candidate point as the score (lower loss $\Rightarrow$ more likely member); (iv) influence-function MIAs (Cohen & Giryes, 2024), which use self-influence magnitude as the score; (v) LiRA (Carlini et al., 2022), which computes a per-example log-likelihood ratio between member and non-member *loss* distributions estimated via shadow models; (vi) model inversion (ModInv) and gradient inversion (GradInv) (Fredrikson et al., 2015; Zhu et al., 2019; Geiping et al., 2020), which do not output a membership score but optimize a reconstruction objective and evaluate privacy via reconstruction quality; and (vii) property inference (Ateniese et al., 2015; Ganju et al., 2018), which trains a meta-classifier on model outputs. Comprehensive evaluations show that attack outcomes depend on datasets, architectures, and attack strategy (Song & Mittal, 2021). Our framework does not instantiate any attack; instead, GNQ quantifies the *intrinsic* privacy risks of the training algorithm itself, yielding a theoretically-grounded, attack-agnostic auditing.

**Theoretical Attack-based Privacy Auditing.** Although, some efforts exist to make attack-based auditing provide some theoretical guarantees, often these efforts are limited to specific model archi-tectures. For instance, (Murakonda et al., 2021; Ye et al., 2022) provide theoretical analyses of MIA and rely on specific models like Bayesian networks. Similarly, Wang et al. (2023) examines GradInv restricting its scope to fully connected layers. The derivation of GNQ is general, and does not make any assumption on the model architecture.

**Auditing Differential Privacy.** Differential privacy formalizes $(\varepsilon, \delta)$-style guarantees for randomized algorithms (Dwork et al., 2006; Dwork & Roth, 2014). In deep learning, these guarantees are commonly obtained with DP-SGD (Abadi et al., 2016). Recent auditing methods empirically check whether implementations achieve their claimed privacy levels (Jagielski et al., 2020; Steinke et al., 2023; Nasr et al., 2023). Program-logical frameworks verify DP guarantees via proof systems (Barthe et al., 2012; Zhang & Kifer, 2017); software such as OpenDP operationalize DP mechanisms (OpenDP, 2025). By contrast, GNQ quantifies the privacy risks of trained models regardless of whether DP was used, rather than proving DP guarantees for specific randomized mechanisms.

**Machine Unlearning.** Machine unlearning studies removing a point's influence and verifying deletion (Bourtoule et al., 2021; Sommer et al., 2022; Warnecke et al., 2023). GNQ is a per-iteration, per-example risk score that encapsulates how the algorithmic components of the training algorithm shape each datapoint's risk. Hence, it can guide risk-aware, training-time unlearning and serves as a built-in metric to audit the resulting change in risk.

## 3 Gradient Uniqueness

In this Section, we present the central result of the paper: that "gradient uniqueness" (or GNQ)—which we define subsequently—can be used to determine the level of disclosure associated with any datapoint during execution of a mini-batch gradient descent algorithm, regardless of the model. To justify the use of GNQ, in the Appendix of the paper we argue mathematically that the amount of information available to *any* attacker, regardless of the attack strategy, in the disclosure of a model learned is bounded by a function that monotonically increases with GNQ. Thus, any defense that reduces the maximum GNQ tends to reduce the success of any attack.

Assume a public dataset $\mathcal{D} = \{d_j\}_{j=1}^N$ from which the private dataset $\mathcal{D}_t$ is sampled uniformly at random without replacement. The sampling distribution serves as an empirical prior on random variable $\mathcal{D}_t$. While an adversary cannot see $\mathcal{D}_t$, the adversary's goal is to use observable information to infer whether $d_j$ is in $\mathcal{D}_t$; if done with high certainty, the adversary is successful. While determining whether $d_j$ is in $\mathcal{D}_t$ is just one type of attack (a so-called membership inference attack (Shokri et al., 2017)), safety from membership inference under a public dataset serves as a proxy for safety from other attacks, in the sense that if an attacker cannot determine if $d_j$ was used, the disclosure regarding $d_j$ was limited.

Assume that $\mathcal{D}_t$ is used to train some arbitrary model using the classical, mini-batch gradient descent (Alg. 1). The adversary uses some attack strategy $\mathcal{F}$ to analyze the learned parameters $\theta_{N_r}$ to determine whether some $d_j$ is in $\mathcal{D}_t$. We consider a worst-case (white-box) setting in which the model architecture and the released final parameters $\theta_{N_r}$ are known to the adversary.

---

**Algorithm 1** Mini-Batch Stochastic Gradient Descent (SGD)

---

**Input:** Full dataset $\mathcal{D} = \{d_j\}_{j=1}^N$, parameterized model $h_\theta$ with parameter vector $\theta \in \mathbb{R}^{N_p}$, point-wise loss function $\ell[\theta, d]$, learning rate $\eta$, batch size $B$, number of training iterations $N_r$
**Output:** Optimized (final) model parameters $\theta_{N_r}$.
Initialize model parameters to $\theta_0$.
Sample a training set $\mathcal{D}_t$ from $\mathcal{D}$ according to sampling distribution $\mathcal{D}_t \sim Train(\mathcal{D})$.
Sample mini-batches $\{\mathcal{B}_i\}_{i=0}^{N_r-1}$ from $\mathcal{D}_t$ according to sampling distribution $\{\mathcal{B}_i\} \sim Batch(\mathcal{D}_t)$.
**for** $i = 0$ **to** $N_r - 1$ **do**
$\quad \hat{g}_i = \frac{1}{B} \sum_{d_j \in \mathcal{B}_i} \nabla_\theta[\ell[\theta_i, d_j]]$
$\quad \theta_{i+1} = \theta_i - \eta \cdot \hat{g}_i$
**end**

---

We are now ready to define gradient uniqueness:

**Definition 1.** Consider training batch $i$. The *gradient uniqueness* of datapoint $d_j$ with respect to batch $i$ is given by:

$$GNQ_{ij} = g_{ij}^\top S^+ g_{ij} \tag{1}$$

where $S = \sum_{\substack{k=1 \\ k \neq j}}^N g_{ik} g_{ik}^\top \in \mathbb{R}^{N_p \times N_p}$ and $S^+$ denotes the Moore–Penrose pseudoinverse of $S$ and $g_{ij} = \nabla_\theta[\ell[\theta_i, d_j]] \in \mathbb{R}^{N_p}$. Note that $S^+ = S^{-1}$ if $S$ is invertible.

Then, the central result of the paper states that this quantity bounds the amount of information that any adversary can glean about $d_j$'s membership in $\mathcal{D}_t$ by examining $\theta_{N_r}$ using $\mathcal{F}$:

**Gradient uniqueness as an upper bound on disclosure (informal).** *The amount of information (measured in bits) extracted by an adversary attempting to determine whether $d_j \in \mathcal{D}_t$ via examination of $\theta_{N_r}$ using any attack mechanism $\mathcal{F}$ is upper-bounded by a function that increases monotonically with $\sum_{i=1}^{N_r-1} GNQ_{ij}$.*

A formal version of these results is given in Appendix A.

## 4  WHAT DOES GRADIENT UNIQUENESS MEASURE?

To show how GNQ measures privacy risk during gradient descent and to give some intuition behind the metric, we consider a simple example—a 2D linear regression model with the squared loss function (Fig. 1 a). The goal is to quantitatively rank the datapoints from the highest risk to the lowest risk.

The computation of GNQ can be geometrically represented as the construction of an ellipse summarizing the gradients; $GNQ_{ij}$ is the extent to which the gradient associated with $d_j$ is an outlier with respect to this elipse, as shown in Fig. 1 b. In this figure, we plot the gradients of each of the seven training points, and the associated elipses. Note that in the definition of $GNQ_{ij}$, the matrix $S$ (and the resulting elipse) is constructed using all the datapoints, except the point $d_j$ for which $GNQ_{ij}$ is computed. Thus there are two ellipses in Fig. 1 b: the blue ellipse is for the case when point 7 is excluded to compute $GNQ_{i7}$, while the red ellipse is for the case when point 7 is included while excluding one of the other six points to compute $GNQ_{ik}$ where $k \in \{1, \cdots, 6\}$. Datapoint 7 has a very high GNQ value because it falls outside of the blue ellipse, while all other datapoints have a low GNQ value, as they fall inside of the red ellipse.

Intuitively, as shown in Fig. 2, the gradients associated with points 1-6 want to rotate the regression line counter-clockwise, centered roughly on datapoint 5—whereas datapoint 7 is doing exactly the opposite, hence the high value for $GNQ_{i7}$.

**GNQ-based auditing vs attack-based auditing.** GNQ immediately suggests an auditing-based privacy scheme: monitor $GNQ_{ij}$ for all training points, and if $GNQ_{ij}$ is large and $d_j$ is sensitive, take action, such as dropping $d_j$ from the dataset, or unlearning $d_j$.

There are other auditing-based methods, the most well-known of which is MIA, which is an attack mechanism (see Sec. 2). MIA attempts to infer whether a given datapoint $d_j$ was part of the training dataset. This goal is typically achieved by computing a membership score $\mathcal{M}[\theta_{N_r}, d_j]$ and comparing it against a decision threshold $\tau$. Often the membership score is (or a function of) the model's loss on the target datapoint, while the decision threshold is a global threshold used for all datapoints determined either based on a heuristic (members ($d_j \in \mathcal{D}_t$) tend to have lower loss, while non-members ($d_j \notin \mathcal{D}_t$) tend to incur higher loss) or obtained by training shadow models.

In general, heuristic-based methods such as MIA produce very different results from GNQ. Consider Fig. 3. In gradient space, the global loss threshold $\ell = \tau$ appears as a horizontal strip bounded by the two lines $\partial \ell / \partial b = \pm \sqrt{2\tau}$. Points inside the strip $\left( |\partial \ell / \partial b| \leq \sqrt{2\tau} \right)$ are classified as members in the training set; points outside are non-members. We consider two possible thresholds $\tau_1$ and $\tau_2$. If we use $\tau_1$, points 2, 3, and 5 are inside the $\tau_1$-strip, so they will be ranked as high risk points, which does not match the GNQ ranking. Moreover, points 1, 4, 6 and 7 are outside the $\tau_1$-strip, so they will be classified safe points; this matches the GNQ ranking regarding points 1, 4, and 6, however it misranks point 7; the most crucial point in terms of privacy risk according to GNQ. Using $\tau_2$, points 2, 3, 5 and 7 are inside the $\tau_2$-strip, so they will be ranked as high risk points—which does not match the GNQ ranking except for point 7. Moreover, points 1, 4 and 6 are outside the $\tau_2$-strip, so they will be wrongly classified as low-disclosure points.

Further, any loss-based ranking depends solely on the residual (vertical distance to the fit), so points 2, 3, and 5—having smaller residuals (lower loss) than point 7—are necessarily ranked as higher risk for any choice of threshold. By contrast, $GNQ_{ij} = g_{ij}^\top S^{-1} g_{ij}$, uses the full gradient geometry: it accounts for (1) the residual $r$, (2) the feature vector (e.g., $[x, 1]$ in our 2D example)—both via the gradient formula $g = -r[x, 1]$—and (3) inter-example correlations via $S^{-1}$.

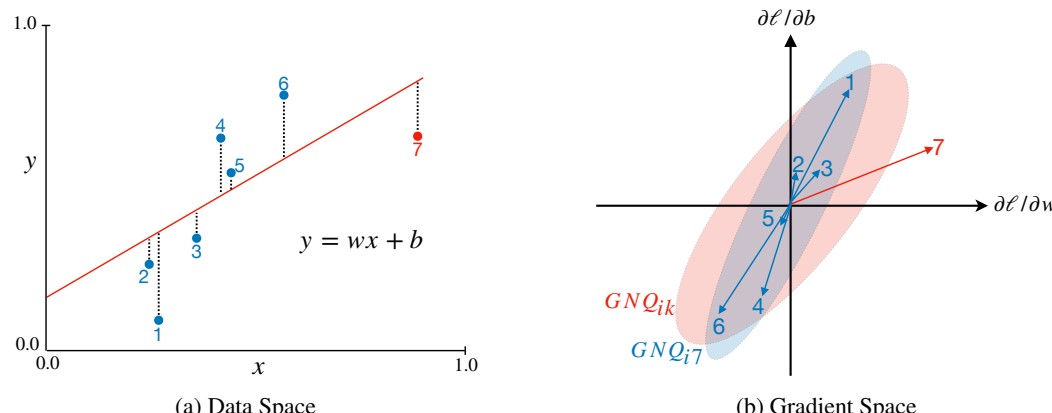

Figure 1: GNQ-based privacy auditing.

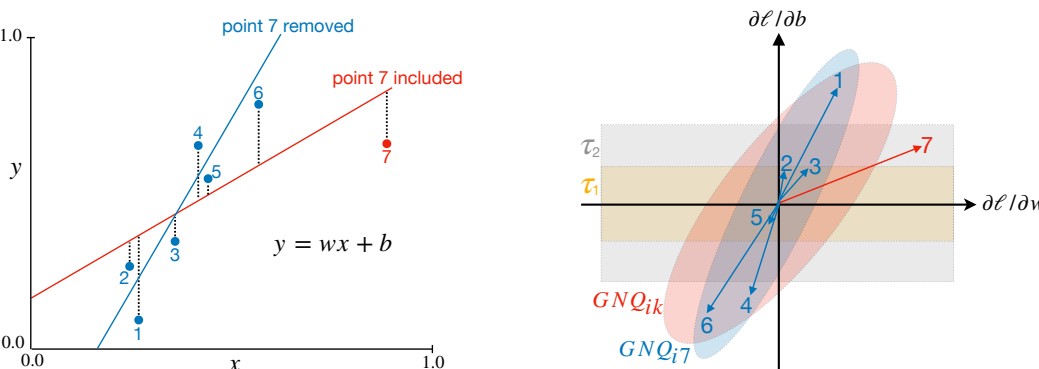

Figure 2: Accuracy of GNQ-based auditing.          Figure 3: GNQ vs. attack-based auditing.

## 5   A GNQ-BASED DEFENSE

While the central result of the paper is the definition and derivation of GNQ, we do wish to show that GNQ can be used to develop a simple defense; we leave other GNQ-based defenses to future work. The simple defense we consider consists of three steps: (i) train and rank datapoints within a dataset from the highest risk to the lowest risk using GNQ, (ii) remove the highest-risk points, and (iii) re-train using the filtered dataset.

For a smaller model with a limited number of parameters, the implementation is very straightforward. For a larger foundation model, implementing this simple GNQ-based defense requires some care, for several reasons. Most critical is the fact that the size of the matrix $S$ scales quadratically with the number of model parameters, and inverting it is likely infeasible for a modern, large, model.

We can reduce the number of parameters that we have to consider by using a smaller, proxy model. That is, if the goal is to filter points before pre-training a 500B parameter model over a sensitive dataset, we can compute GNQ once at the end of a single training epoch using a 3B model in the same family. If this is still too expensive—and it is probably not reasonable to invert a 3B $\times$ 3B matrix—we can rely on approximations of $S^+$. The simplest is to compute only the diagonal of $S$ and assume the off-diagonal elements are zero. It is this approximation we use in our experiments.

A final way to speed the computation is to only compute $S$ using $B$ gradients instead of $N$ gradients. That is, when we compute the gradients over $\mathcal{B}_i$ during gradient descent, for $d_j \in \mathcal{B}_i$, we compute $GNQ_{ij}$ using $S = \sum_{\substack{d_k \in \mathcal{B}_i \\ k \neq j}} g_{ik} g_{ik}^\top$ instead of $S = \sum_{\substack{k=1 \\ k \neq j}}^{N} g_{ik} g_{ik}^\top$. This significantly reduces the number of gradient vectors that need to be memorized.

## 6 EXPERIMENTS

We evaluate on six datasets—MNIST (LeCun et al., 1998), CIFAR-10/100 (Krizhevsky, 2009), the AT&T Database of Faces (Samaria & Harter, 1994), Tiny ImageNet (Le & Yang, 2015; Deng et al., 2009), and IMDB (Maas et al., 2011)—using four model families: an MLP classifier, a basic CNN, ResNet (He et al., 2016) for vision, and a BERT-based classifier for text (Devlin et al., 2019). Models trained with Adam optimizer (Kingma & Ba, 2015).

### 6.1 GNQ-BASED DEFENSE

We begin by evaluating the utility of the simple, GNQ-based defense in the previous section.

**MIA Experiment.** We begin with a series of experiments that use a GNQ calculation—using the diagonalized $S$ approximation—on a small version of a model in the same family (as described in Sec. 5), to filter out high-risk points. Our goal is to examine the privacy/model utility tradeoff obtained using this defense. Privacy is measured using the success of an MIA attack on the resulting model, to identify the points that were left out of the training set.

Specifically, we compute GNQ using a tiny CNN on CIFAR10, a tiny CNN on CIFAR100, a tiny ResNet on Tiny ImageNet, and a tiny BERT-based classifier on IMDB. Each tiny model is trained for a single epoch; we compute GNQ once at the end of that epochs. For each model-dataset pair, we remove the top-$p\%$ highest-risk points according to GNQ and retrain using the larger model. We also consider a non-private model ("Baseline") and models trained with DP-SGD at $\epsilon \in \{2, 8, 512\}$.

*Results.* Table 1 shows both the AUC ROC obtained using a MIA (Salem et al., 2018), and the test accuracy obtained when using the learned model, after private training (or non-private training in the case of the baseline). For example, for CFAR100, we report how accurate is the resulting model in classifying test images.

These results show that it is possible to use the GNQ-based defense to push the attack AUC ROC roughly down to the level of random guessing, while obtaining significantly higher accuracy than what is obtained using DP-SGD.

**ModInv Experiment.** We train an MLP on AT&T Database of Faces. We rank examples by GNQ, and remove the single highest-GNQ-scored example (from class 6, the most vulnerable class by mean score), and retrain. We then apply the class-level ModInv attack of (Fredrikson et al., 2015) to class 6 before and after filtering.

*Results.* On the original dataset (95.31% validation accuracy), high-GNQ content is clearly reconstructed; after removing just one example, the filtered model (94.15% validation accuracy) substantially degrades reconstruction for the same class (Fig. 4).

### 6.2 CAN GRADIENT UNIQUENESS EXPLAIN WHAT'S OBSERVED IN PRACTICE?

In this section, we empirically validate the ability of GNQ to explain/predict empirical observations regarding the success rate of attacks, in terms of recovering a point or determining whether it was used in training, as well as observations regarding the impact of standard, algorithmic components of the SGD algorithm (batch size, learning rate, etc.) on the vulnerability of a learned model.

**GNQ as an explanation for MIA success.** We train four model–dataset pairs: a CNN on CIFAR10, a CNN on CIFAR100, a ResNet on Tiny ImageNet, and a BERT-based classifier on IMDB. For each trained model, we compute GNQ at the end of every epoch and summarize each example using the total GNQ. We then attack the learned model with the MIA of Salem et al. (2018) and record, per example, whether the attack succeeded. The goal is to see if there is a strong relationship between attack success and GNQ. To visualize the relation between our score and attack outcomes, we bin examples by GNQ on a logarithmic grid; the grey bars in Fig. 5 show the number of examples per bin. For each bin we compute the mean MIA success rate (blue curve).

*Results.* We find that across all models and datasets, the success rate increases with GNQ; examples with larger GNQ values are precisely those on which the attack is more likely to succeed.

**GNQ as an explanation for ModInv success.** We train two model-dataset pairs: an MLP on AT&T Database of Faces and a ResNet on MNIST. We then apply the class-level ModInv attack of

Table 1: Comparing GNQ-filtered SGD to DP-SGD for membership inference defense.

| Dataset | Model | Setting | AUC ROC | Test Accuracy |
|---|---|---|---|---|
| CIFAR10 | ResNet | Baseline | 0.7294 | 80.80% |
| | | Top-1% GNQ Removed | 0.6896 | 77.98% |
| | | Top-5% GNQ Removed | 0.6002 | 73.66% |
| | | Top-10% GNQ Removed | **0.5122** | **71.33%** |
| | | DP-SGD ($\epsilon = 2$) | 0.5008 | 41.83% |
| | | DP-SGD ($\epsilon = 8$) | 0.4998 | 47.18% |
| | | DP-SGD ($\epsilon = 512$) | 0.5030 | 55.99% |
| CIFAR100 | ResNet | Baseline | 0.8752 | 49.58% |
| | | Top-1% GNQ Removed | 0.8213 | 46.19% |
| | | Top-5% GNQ Removed | 0.8010 | 43.00% |
| | | Top-10% GNQ Removed | 0.6959 | 41.17% |
| | | Top-15% GNQ Removed | 0.6589 | 37.39% |
| | | Top-20% GNQ Removed | **0.5137** | **34.92%** |
| | | DP-SGD ($\epsilon = 2$) | 0.5015 | 6.83% |
| | | DP-SGD ($\epsilon = 8$) | 0.5008 | 9.14% |
| | | DP-SGD ($\epsilon = 512$) | 0.5000 | 18.39% |
| Tiny ImageNet | ResNet | Baseline | 0.9542 | 38.67% |
| | | Top-1% GNQ Removed | 0.9042 | 35.24% |
| | | Top-5% GNQ Removed | 0.8152 | 32.14% |
| | | Top-10% GNQ Removed | 0.7387 | 32.39% |
| | | Top-15% GNQ Removed | 0.6229 | 29.18% |
| | | Top-20% GNQ Removed | **0.5159** | **25.84%** |
| | | DP-SGD ($\epsilon = 2$) | 0.4961 | 2.15% |
| | | DP-SGD ($\epsilon = 8$) | 0.4981 | 3.73% |
| | | DP-SGD ($\epsilon = 512$) | 0.5044 | 9.87% |
| IMDB | BERT | Baseline | 0.7444 | 81.98% |
| | | Top-0.5% GNQ Removed | 0.5852 | 79.30% |
| | | Top-1% GNQ Removed | 0.6908 | 83.07% |
| | | Top-5% GNQ Removed | **0.4972** | **80.48%** |
| | | DP-SGD ($\epsilon = 2$) | 0.4940 | 50.58% |
| | | DP-SGD ($\epsilon = 8$) | 0.4966 | 51.23% |

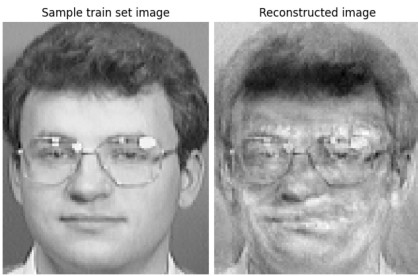

(a) Before Filtration

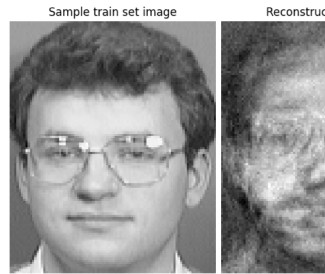

(b) After Filtration

Figure 4: Image reconstruction after removing the highest-ranked image by GNQ.

(Fredrikson et al., 2015) on these trained models, to reconstruct the examples used during training. For each reconstructed image from a target class, we computed the minimum feature distance to any training example from that class. Feature distances were measured as the squared $\ell_2$ norm between activations in the penultimate layer. A small feature distance corresponds to a successful attack, as the recovered image matches others in the class. Fig. 6 shows the relationship between GNQ and feature distance for each class (blue) across different models and datasets. The reported GNQ for a class is the sum of GNQ values of the individual examples, accumulated over all training iterations.

*Results.* The observed downward trend indicates a strong negative correlation between GNQ and feature distance, with Pearson correlation coefficients of $r = -0.90$ and $r = -0.96$. This trend

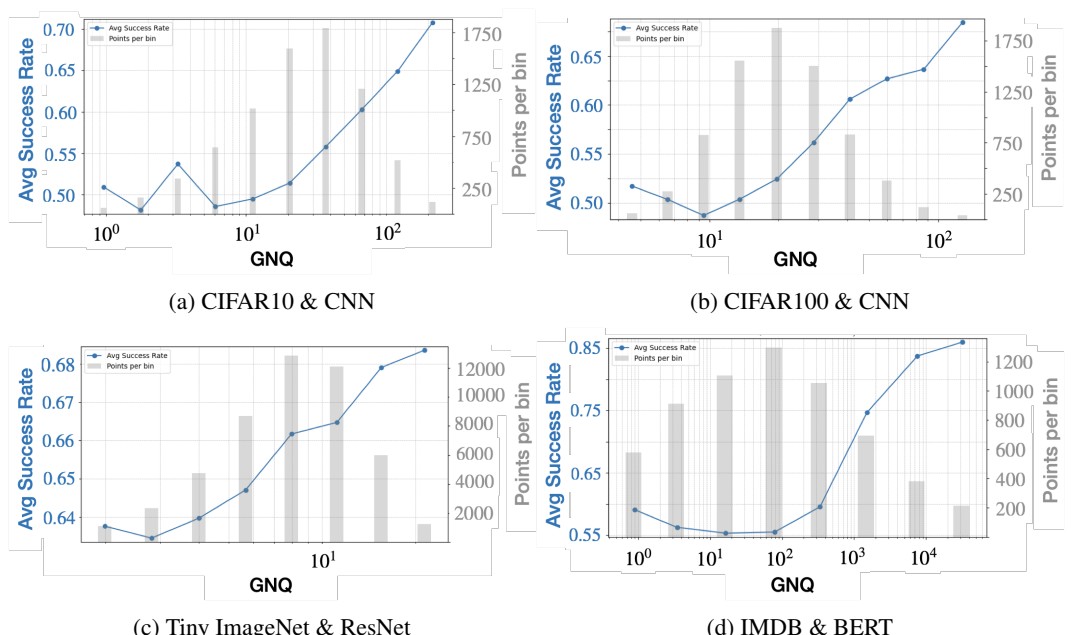

(a) CIFAR10 & CNN

(b) CIFAR100 & CNN

(c) Tiny ImageNet & ResNet

(d) IMDB & BERT

Figure 5: Attack success rate vs. GNQ.

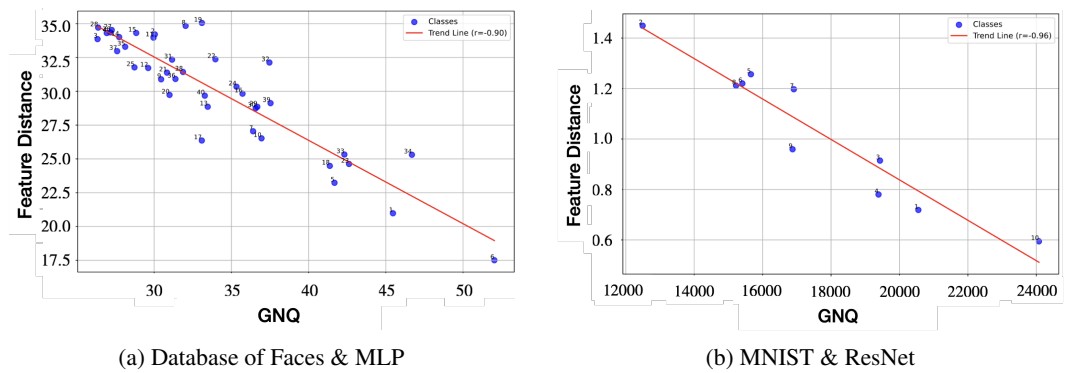

(a) Database of Faces & MLP

(b) MNIST & ResNet

Figure 6: Feature distance vs. GNQ. Each blue circle represents a class.

suggests that classes with higher GNQ values tend to have smaller feature distances, meaning that reconstructed images are closer to real training samples. In Appendix B, we identified two classes of interest from the Database of Faces dataset: the class with the lowest GNQ (class 28) and the class with the highest GNQ (class 6).

**GNQ as an explanation for SGD parameters and privacy.** Finally, we demonstrate how GNQ can explain/predict the relationship between the privacy inherent to an SGD-style algorithm and the parameters used or chosen during training (dataset size, number of iterations, etc.) using the CIFAR-10 dataset with a CNN-based classifier. We consider one parameter setting at a time. For each setting, GNQ is computed for each training point at the end of each epoch, and we consider the maximum observed during training. The mean of the maximum values is used as a metric for the vulnerability of a particular parameter setting.

*Results.* We consider five parameters: the number of iterations (Fig. 7 a), the dataset size (Fig. 7 b), the model size (Fig. 7 c), the batch size (Fig. 7 d), and learning rate (Fig. 7 e). In each case, we plot the average-max GNQ value, as well as the MIA attack success rate (Salem et al., 2018) (MIA attack success is measured via AUC ROC). The results show that GNQ predicts, almost perfectly, how attack success will vary with various training parameters. However, it is important to note that these trends do not necessarily hold over all datasets and all attacks. While Fig. 7 a shows that

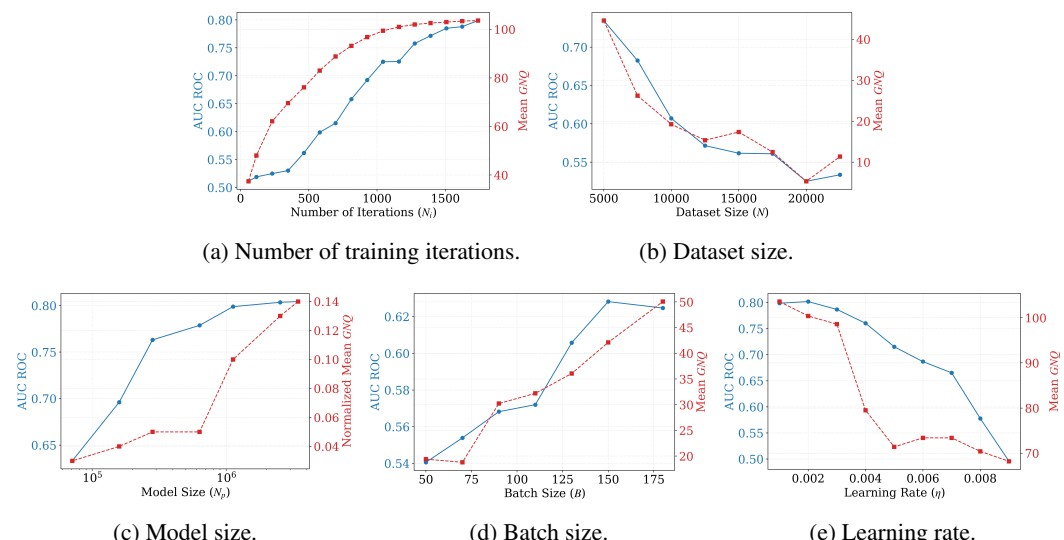

(a) Number of training iterations.

(b) Dataset size.

(c) Model size.

(d) Batch size.

(e) Learning rate.

Figure 7: Relationship between various learning/dataset parameters, mean GNQ, and attack success (MIA attack, measured using AUC ROC) in the CFAR-10 dataset. Plots show mean GNQ (right axis, red) and attack AUC ROC (left axis, blue).

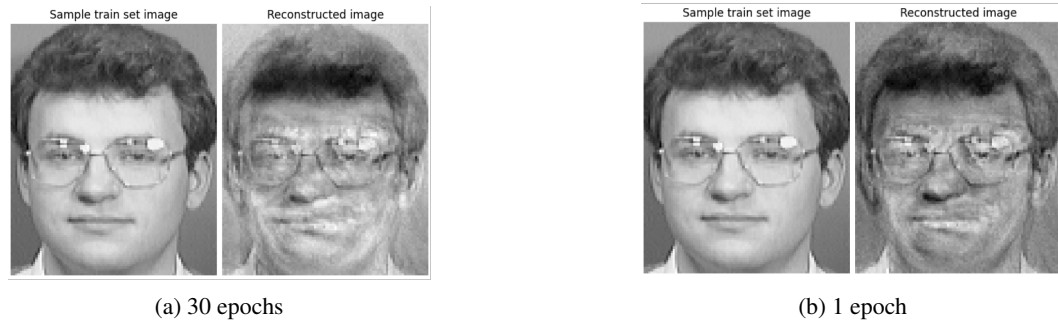

(a) 30 epochs

(b) 1 epoch

Figure 8: ModInv applied to the highest GNQ class in the Faces dataset. Reducing the number of epochs down to one fails to protect this class.

reducing the number of epochs affords some protection, it does not always afford this protection, and GNQ is able to determine vulnerable training points in an attack-agnostic way. We show, in Fig. 8, that even training for only a single epoch on the Faces dataset fails to protect high-GNQ classes from a reconstruction attack. Importantly, GNQ captures this vulnerability while being attack-agnostic.

# 7 CONCLUSION

We have derived a quantity called *gradient uniqueness* (GNQ) that increases monotonically with a bound on the information disclosure during mini-batch gradient descent. We have showed that indeed, GNQ strongly correlates with attack success. Future work should explore more sophisticated defenses based on GNQ. For example, one need not simply censor an entire datapoint with high GNQ, but could add a bit of noise to the gradient of a high-risk point, even adding noise only to certain dimensions of the gradient. GNQ may also serve emerging areas such as machine unlearning.

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

## A  GRADIENT UNIQUENESS AS A MEASURE OF DATA PRIVACY

### A.1  OVERVIEW

In this section, we argue that gradient uniqueness is a practical and effective measure for quantifying data privacy. It is straightforward to construct adversarial scenarios where the inclusion of a single datapoint in a training set can be inferred from a trained model. Consequently, it is impossible to formulate a general theorem that guarantees protection against such membership inference attacks in all cases.

However, there is a consensus among practitioners that such worst-case scenarios are unlikely to occur in practice. Our objective is to bridge the gap between this practical intuition and theoretical guarantees. We show that under common practical assumptions, the gradient uniqueness measure rigorously quantifies the risk of an adversary successfully inferring the membership of a specific datapoint.

Our argument is grounded in information theory (Cover & Thomas, 2006). Our result shows that under some practical assumptions, the randomness in the mini-batch SGD (Algorithm 1) provides some level of protection for the membership of the datapoint $d_j$ in the training dataset $\mathcal{D}_t$ denoted by $T_j$. It is well-known that if the mutual information between a target variable, say $T_j$, and an observable variable is small, then no algorithm can predict $T_j$ with a success probability significantly better than a random guess (i.e., 50%). We formally demonstrate that the mutual information between the membership variable $T_j$ and the released model is *upper-bounded by an increasing function of gradient uniqueness*. This result provides a firm theoretical justification for using gradient uniqueness as a proxy for the information leakage associated with a particular datapoint.

**Problem setup:**  Before stating our core argument, we recall the setup of our problem and Algorithm 1. To simplify our analysis, we define the sampling distributions $Train()$ and $Batch()$ as follows. For each datapoint $d_j$, let $T_j \sim \mathrm{Ber}(N_t/N)$ control whether $d_j$ is included in $\mathcal{D}_t$. Assume $N_t = N/2$. For training batch $i$, let $M_{i,j} \sim \mathrm{Ber}(B/N_t)$ be a random variable contorlling whether $d_j$ is included in the mini-batch. In particular, we include the datapoint $d_j$ in the batch if and only if $T_j \cdot M_{i,j} = 1$. All these Bernoulli random variables are drawn independently.

We denote the gradient of the loss function at iteration $i$, evaluated at datapoint $d_j$, by

$$g_{i,j} := \nabla_\theta \ell(\theta_i, d_j).$$

The mini-batch gradient is defined as the average of these gradients, normalized by $B$ (the expected batch size) rather than the realized batch size:

$$\hat{g}_i := \frac{1}{B} \sum_{j=1}^{N} T_j \cdot M_{i,j} \cdot g_{i,j} = \frac{1}{B} \sum_{j=1}^{N} T_j \cdot M_{i,j} \cdot \nabla_\theta \ell(\theta_i, d_j).$$

The SGD algorithm runs in $N_r$ iterations with the following update rule:

$$\theta_{i+1} = \theta_i - \eta \cdot \hat{g}_i,$$

where $\eta$ is a fixed step size. The initial model $\theta_0$ is chosen independently of the dataset; equivalently, for all $T_j$, we have $I[T_j; \theta_0] = 0$.

We emphasize that there is no randomness in the overall dataset $\mathcal{D}$; that is the set of all possible datapoints is known to everyone. The training set, however, is a random subset of $\mathcal{D}$. The fact that a particular datapoint is used or not is a quantity that we would like to protect.

**Our assumptions:**  Our subsequent theoretical development hinges on two main assumptions that do not hold in general. However, these assumptions are motivated by empirical observations about the SGD algorithm.

- Gradients are often empirically observed to resemble Gaussian distributions (Panigrahi et al., 2019; Millidge & Winsor, 2023). Motivated by this observation, we assume that the entropy of the batch gradients can be approximated by that of a multivariate normal distribution with the same covariance structure. For a formal statement, see Equation 3.

- We assume that in each iteration the gradients measured at individual datapoints $g_{i,j}$ reside within the same subspace spanned by the rest of the data $\{g_{i,j'}\}_{j \neq j'}$. In other words, there are no outlier points whose gradients drive the model in directions unique to themselves.

## A.2 GRADIENT UNIQUENESS AS AN UPPER BOUND ON DISCLOSURE

In this section, we utilize the assumptions of the prior section to argue that gradient uniqueness is a practical bound on information disclosure. Various theorems and lemmas used in our argument are given in Sec. A.3.

As stated, our goal is to define a proxy that captures an attacker's ability to infer $T_j$ from the final model $\theta_{N_r}$. Prior to training, an adversary only knows that $T_j \sim \text{Ber}(B/N_t)$, yielding a trivial baseline guess. The mutual information then quantifies how much this baseline can be improved when the adversary observes $\theta_{N_r}$.

We begin by decomposing the mutual information between $T_j$ and the final model $\theta_{N_r}$ and relate it to the entropy of the mini-batch gradients using information-theoretic tools. In particular, we will rely on the following theorem. The proof of this theorem is provided in Sec. A.3.

**Theorem A.1.** *In mini-batch SGD (Algorithm 1), for any $j \in [N]$ and corresponding datapoint $d_j \in \mathcal{D}$, the mutual information between its training membership indicator $T_j \in \{0,1\}$ and the estimate $\hat{T}_j := \mathcal{F}(\theta_{N_r}, j)$ produced by an attacker $\mathcal{F}$ is bounded by:*

$$
I\left[T_j; \mathcal{F}(\theta_{N_r}, j)\right]
$$
$$
\leq \sum_{i=1}^{N_r - 1} H[\hat{g}_i \mid \theta_i] - H[\hat{g}_i \mid \theta_i, T_j = 0] - \frac{N_t}{N} \cdot \left( H[\hat{g}_i \mid \theta_i, T_j = 1] - H[\hat{g}_i \mid \theta_i, T_j = 0] \right).
$$
(2)

Given this theorem, we now use our assumption that the distribution of $\hat{g}_i$ resembles a (possibly degenerate) multivariate normal distribution to approximate its entropy as:

$$
H[\hat{g}_i \mid \theta_i] \approx \tfrac{1}{2} \log\left((2\pi e)^r \, \text{pdet}(\Sigma)\right),
$$
(3)

where $\Sigma$ denotes the covariance matrix of $\hat{g}_i$ conditioned on $\theta_i$, $\text{pdet}(\Sigma)$ is the pseudo-determinant of $\Sigma$ (the product of its nonzero eigenvalues), and $r = \text{rank}(\Sigma)$. Similarly, we define $\Sigma^{(j,0)}$ (resp. $\Sigma^{(j,1)}$) as the covariance matrix of $\hat{g}_i$ conditioned on $\theta_i$ and $T_j = 0$ (resp. $T_j = 1$). We derive these covariance matrices explicitly in Sec. A.4. As we show, the matrices are given by:

$$
\Sigma = \frac{1}{B\,N} \cdot \left(1 - \frac{B}{N}\right) \sum_{j=1}^{N} g_{i,j}\, g_{i,j}^\top
$$

$$
\Sigma^{(j,0)} = \frac{1}{B\,N} \cdot \left(1 - \frac{B}{N}\right) \sum_{j' \neq j} g_{i,j'}\, g_{i,j'}^\top
$$

$$
\Sigma^{(j,1)} = \frac{1}{BN_t} \cdot \left(1 - \frac{B}{N_t}\right) \cdot g_{i,j}\, g_{i,j}^\top + \frac{1}{B\,N} \cdot \left(1 - \frac{B}{N}\right) \sum_{j' \neq j} g_{i,j'}\, g_{i,j'}^\top.
$$

It is not hard to see that $\Sigma$ and $\Sigma^{(j,1)}$ are rank-one perturbation of $\Sigma^{(j,0)}$. In particular, we have:

$$
\Sigma = \Sigma^{(j,0)} + \underbrace{\frac{1}{B\,N} \cdot \left(1 - \frac{B}{N}\right)}_{c_1^2 :=} \cdot g_{i,j}\, g_{i,j}^\top
$$

$$
\Sigma^{(j,1)} = \Sigma^{(j,0)} + \underbrace{\frac{1}{BN_t} \cdot \left(1 - \frac{B}{N_t}\right)}_{c_2^2 :=} \cdot g_{i,j}\, g_{i,j}^\top.
$$

Let us define $q = c_1 g_{i,j}$ and $q' = c_2 g_{i,j}$ where $c_1$ and $c_2$ are the constant we have above. Note that earlier we assumed that $g_{i,j}$ belongs to the span of $\{g_{i,j'}\}_{j \neq j'}$. In Lemma A.2, we show that $q$ and $q'$ must have belong to the range of $\Sigma^{(j,0)}$. Furthermore, in Lemma A.3 we show that adding $qq^\top$ (or $q'q'^\top$) does not change rank of $\Sigma^{(j,0)}$. Our assumptions lead to a clean approximation of the right hand side of Equation 2. In particular, we have the following bound via Theorem A.1:

$$I\left[T_j; \mathcal{F}(\theta_{N_r}, j)\right]$$

$$\leq \sum_{i=1}^{N_r-1} H[\hat{g}_i \mid \theta_i] - H[\hat{g}_i \mid \theta_i, T_j = 0] - \frac{N_t}{N} \cdot \left(H[\hat{g}_i \mid \theta_i, T_j = 1] - H[\hat{g}_i \mid \theta_i, T_j = 0]\right)$$

$$\approx \sum_{i=1}^{N_r-1} \frac{1}{2} \left(\log\left(\frac{\mathrm{pdet}\,(\Sigma)}{\mathrm{pdet}\left(\Sigma^{(j,0)}\right)}\right) - \frac{N_t}{N} \log\left(\frac{\mathrm{pdet}\left(\Sigma^{(j,1)}\right)}{\mathrm{pdet}\left(\Sigma^{(j,0)}\right)}\right)\right)$$

$$= \sum_{i=1}^{N_r-1} \frac{1}{2} \left(\log\left(1 + c_1^2 g_{i,j}^\top \left(\Sigma^{(j,0)}\right)^+ g_{i,j}\right) - \frac{N_t}{N} \log\left(1 + c_2^2 g_{i,j}^\top \left(\Sigma^{(j,0)}\right)^+ g_{i,j}\right)\right)$$

$$= \sum_{i=1}^{N_r-1} \frac{1}{2} \left(\log\left(\frac{1 + c_1^2 g_{i,j}^\top \left(\Sigma^{(j,0)}\right)^+ g_{i,j}}{\left(1 + c_2^2 g_{i,j}^\top \left(\Sigma^{(j,0)}\right)^+ g_{i,j}\right)^{N_t/N}}\right)\right).$$

Assuming that $N_t = N/2$, what we have in the last line is an increasing function of $x := g_{i,j}^\top \left(\Sigma^{(j,0)}\right)^+ g_{i,j}$. We prove this by showing that

$$f(x) := \frac{1 + c_1^2 x}{\sqrt{1 + c_2^2 x}}$$

is an increasing function of $x$ in Lemma A.4 for our desired range of parameters. This completes the argument that gradient uniqueness serves as a practical upper bound on the information available to an attacker on whether $T_j = 1$.

## A.3 Proofs of Theorems and Lemmas

**Theorem A.1.** *In mini-batch SGD (Algorithm 1), for any $j \in [N]$ and corresponding datapoint $d_j \in \mathcal{D}$, the mutual information between its training membership indicator $T_j \in \{0, 1\}$ and the estimate $\hat{T}_j := \mathcal{F}(\theta_{N_r}, j)$ produced by an attacker $\mathcal{F}$ is bounded by:*

$$I\left[T_j; \mathcal{F}(\theta_{N_r}, j)\right]$$

$$\leq \sum_{i=1}^{N_r-1} H[\hat{g}_i \mid \theta_i] - H[\hat{g}_i \mid \theta_i, T_j = 0] - \frac{N_t}{N} \cdot \left(H[\hat{g}_i \mid \theta_i, T_j = 1] - H[\hat{g}_i \mid \theta_i, T_j = 0]\right). \tag{2}$$

*Proof.* Fix $j \in [N]$ for the rest of the proof. Note that using the SGD update rule, the final model $\theta_{N_r}$ is obtained from the second to last model $\theta_{N_r-1}$ together with the gradient in the last batch $\hat{g}_{N_r-1}$. By the data processing inequality, we obtain:

$$I\left[T_j; \mathcal{F}(\theta_{N_r}, j)\right] \leq I[T_j; \theta_{N_r}] \leq I[T_j; \theta_{N_r-1}, \hat{g}_{N_r-1}]$$
$$= I[T_j; \theta_{N_r-1}] + I[T_j; \hat{g}_{N_r-1} \mid \theta_{N_r-1}] \qquad \text{(via chain rule)}$$

By recursively applying this decomposition from step $N_r$ down to step 0, we obtain:

$$I\left[T_j; \mathcal{F}(\theta_{N_r}, j)\right] \leq I[T_j; \theta_0] + \sum_{i=1}^{N_r-1} I[T_j; \hat{g}_i \mid \theta_i].$$

Since the initial model $\theta_0$ is chosen independently of the training data, we have $I[T_j; \theta_0] = 0$. Hence, we obtain:

$$I\left[T_j; \mathcal{F}(\theta_{N_r}, j)\right] \leq \sum_{i=1}^{N_r-1} I[T_j; \hat{g}_i \mid \theta_i]. \tag{4}$$

We now analyze each term in the above summation. By definition of mutual information and the conditional entropy, we have

$$
\begin{aligned}
I[T_j; \hat{g}_i \mid \theta_i] &= H[\hat{g}_i \mid \theta_i] - H[\hat{g}_i \mid \theta_i, T_j] \\
&= H[\hat{g}_i \mid \theta_i] - \mathbb{P}[T_j = 0] \cdot H[\hat{g}_i \mid \theta_i, T_j = 0] - \mathbb{P}[T_j = 1] \cdot H[\hat{g}_i \mid \theta_i, T_j = 1] \\
&= H[\hat{g}_i \mid \theta_i] - \left[1 - \frac{N_t}{N}\right] \cdot H[\hat{g}_i \mid \theta_i, T_j = 0] - \left[\frac{N_t}{N}\right] \cdot H[\hat{g}_i \mid \theta_i, T_j = 1] \\
&= H[\hat{g}_i \mid \theta_i] - H[\hat{g}_i \mid \theta_i, T_j = 0] - \frac{N_t}{N} \cdot \Big( H[\hat{g}_i \mid \theta_i, T_j = 1] - H[\hat{g}_i \mid \theta_i, T_j = 0] \Big)
\end{aligned}
\tag{5}
$$

Here we used that $T_j$ is a Bernoulli random variable with parameter $N_t/N$. Substituting Equation equation 5 into Equation equation 4 yields the stated bound, completing the proof. □

**Lemma A.2.** *Let $x_1, \ldots, x_n \in \mathbb{R}^d$ (not necessarily linearly independent) and define*

$$A = \sum_{i=1}^{n} x_i x_i^\top \in \mathbb{R}^{d \times d}.$$

*Then* $\mathrm{Range}(A) = \mathrm{span}\{x_1, \ldots, x_n\}$. *In particular, each $x_i$ lies in* $\mathrm{Range}(A)$.

*Proof.* First, for any $y \in \mathbb{R}^d$ we have

$$Ay = \sum_{i=1}^{n} x_i(x_i^\top y) \in \mathrm{span}\{x_1, \ldots, x_n\},$$

so $\mathrm{Range}(A) \subseteq \mathrm{span}\{x_1, \ldots, x_n\}$.

For the reverse inclusion, observe that $A$ is symmetric positive semidefinite, and its nullspace is

$$\mathcal{N}(A) = \{\, y \in \mathbb{R}^d : Ay = 0 \,\} = \{\, y \in \mathbb{R}^d : x_i^\top y = 0 \text{ for all } i \,\},$$

since $y^\top A y = \sum_{i=1}^{n}(x_i^\top y)^2 = 0$ iff $x_i^\top y = 0$ for all $i$. Therefore $\mathcal{N}(A) = \mathrm{span}\{x_1, \ldots, x_n\}^\perp$. By the Fundamental Theorem of Linear Algebra,

$$\mathrm{Range}(A) = \mathcal{N}(A)^\perp = \big(\mathrm{span}\{x_1, \ldots, x_n\}^\perp\big)^\perp = \mathrm{span}\{x_1, \ldots, x_n\}.$$

Hence each $x_i \in \mathrm{Range}(A)$. □

**Lemma A.3.** *Let $A \in \mathbb{R}^{n \times n}$ be a symmetric matrix (not necessarily full-rank), and let $q \in \mathrm{range}(A)$. Then*

$$\mathrm{pdet}(A + qq^\top) = \mathrm{pdet}(A)\left(1 + q^\top A^+ q\right),$$

*where $\mathrm{pdet}(\cdot)$ denotes the pseudo-determinant (the product of the nonzero eigenvalues) and $A^+$ is the Moore–Penrose pseudoinverse. Moreover, the rank of $A$ and $A + qq^\top$ are the same.*

*Proof.* Since $A$ is symmetric, by the spectral theorem we may write

$$A = \sum_{i=1}^{r} \lambda_i u_i u_i^\top,$$

where $\lambda_1, \ldots, \lambda_r \neq 0$ are the nonzero eigenvalues of $A$, $u_1, \ldots, u_r$ are the corresponding orthonormal eigenvectors, and the remaining eigenvalues are zero. Equivalently,

$$A = U \begin{bmatrix} \Lambda_r & 0 \\ 0 & 0 \end{bmatrix} U^\top,$$

where $U = [U_r \; U_0]$ is orthogonal, $\Lambda_r = \mathrm{diag}(\lambda_1, \ldots, \lambda_r)$, and $\mathrm{range}(A) = \mathrm{span}(U_r)$.

Since $q \in \mathrm{range}(A)$, we can write $q = U_r y$ for some $y \in \mathbb{R}^r$. Then

$$A + qq^\top = U \begin{bmatrix} \Lambda_r + yy^\top & 0 \\ 0 & 0 \end{bmatrix} U^\top.$$

The nonzero spectrum of $A + qq^\top$ is the nonzero spectrum of $\Lambda_r + yy^\top$.[1] Note that since $\Lambda_r$ is positive definite and $yy^\top$ is positive semi-definite, their sum $\Lambda_r + yy^\top$ is also positive definite. It is therefore full rank and has a non-zero determinant. This representation also implies that the number of orthonormal eigenvectors with nonzero eigenvalues does not change after perturbation, meaning that the ranks of $A$ and $A + qq^\top$ are the same. Moreover, we have:

$$\mathrm{pdet}(A + qq^\top) = \det(\Lambda_r + yy^\top).$$

Applying the matrix determinant lemma implies

$$\det(\Lambda_r + yy^\top) = \det(\Lambda_r)\left(1 + y^\top \Lambda_r^{-1} y\right).$$

Now note that

$$\det(\Lambda_r) = \mathrm{pdet}(A), \qquad y^\top \Lambda_r^{-1} y = q^\top A^+ q,$$

where $A^+ = U_r \Lambda_r^{-1} U_r^\top$ is the Moore–Penrose pseudoinverse of $A$.

Therefore,

$$\mathrm{pdet}(A + qq^\top) = \mathrm{pdet}(A)\left(1 + q^\top A^+ q\right),$$

as claimed. $\qquad\square$

**Lemma A.4.** *Let $c_1, c_2 > 0$ and, for $x \geq 0$, define*

$$f(x) \; := \; \frac{1 + c_1^2 x}{\sqrt{1 + c_2^2 x}}.$$

*If $2c_1^2 > c_2^2$, then $f$ is an increasing function of $x$ on $[0, \infty)$.*

*Proof.* For $x \geq 0$,

$$f(x) = (1 + c_1^2 x)(1 + c_2^2 x)^{-1/2},$$

so by the product/chain rules,

$$f'(x) = c_1^2 (1 + c_2^2 x)^{-1/2} - \tfrac{1}{2}(1 + c_1^2 x) c_2^2 (1 + c_2^2 x)^{-3/2} = \frac{\left(2c_1^2 - c_2^2\right) + c_1^2 c_2^2 \, x}{2(1 + c_2^2 x)^{3/2}}.$$

The denominator is positive for all $x \geq 0$. Under $2c_1^2 > c_2^2$ the numerator is positive for all $x \geq 0$. Hence $f'(x) > 0$ on $[0, \infty)$, so $f$ is an increasing function of $x$ on this interval. $\qquad\square$

### A.4 DERIVATION OF COVARIANCE MATRICES

Recall that we define $\Sigma$ to be the covariance matrix of $\hat{g}_i$ conditioned on $\theta_i$. Our derivation begins by calculating the $(a, b)$ entry of the $\Sigma$, and expanding it. In the following, we use $g_{i,j}^{(a)}$ to denote the $a$th coordinate of the vector $g_{i,j}$.

$$\Sigma_{ab} = \frac{1}{B^2} \cdot \mathrm{cov}\left(\sum_{j=1}^{N} T_j \, M_{i,j} \, g_{i,j}^{(a)}, \sum_{j'=1}^{N} T_{j'} \, M_{i,j'} \, g_{i,j'}^{(b)}\right)$$

By the definition of covariance, this can be written as:

$$\Sigma_{ab} = \frac{1}{B^2} \cdot \left( E\left[\left(\sum_{j=1}^{N} T_j \, M_{i,j} \, g_{i,j}^{(a)}\right)\left(\sum_{j'=1}^{N} T_{j'} \, M_{i,j'} g_{i,j'}^{(b)}\right)\right] - E\left[\sum_{j=1}^{N} T_j \, M_{i,j} \, g_{i,j}^{(a)}\right] E\left[\sum_{j'=1}^{N} T_{j'} \, M_{i,j'} g_{i,j'}^{(b)}\right]\right).$$

---

[1] If $A = UBU^{-1}$ with $U$ invertible, then $w$ is an eigenvector of $B$ with eigenvalue $\lambda$ iff $U^{-1}w$ is an eigenvector of $A$ with the same eigenvalue $\lambda$.

Rearranging the sums and expectations gives:

$$\Sigma_{ab} = \frac{1}{B^2} \cdot \sum_{j,j'=1}^{N} \left( E[T_j \, M_{i,j} T_{j'} \, M_{i,j'}] - E[T_j \, M_{i,j}] E[T_{j'} \, M_{i,j'}] \right) g_{i,j}^{(a)} \, g_{i,j}^{(b)}$$

This simplifies to a sum of covariances:

$$\Sigma_{ab} = \frac{1}{B^2} \cdot \sum_{j,j'=1}^{N} \operatorname{cov}(T_j \, M_{i,j}, T_{j'} \, M_{i,j'}) g_{i,j}^{(a)} \, g_{i,j'}^{(b)}$$

Given that for $j \neq j'$, the terms $T_j \, M_{i,j}$ and $T_{j'} \, M_{i,j'}$ are independent, their covariance is zero. This eliminates the terms where $j \neq j'$, leaving only the terms where $j = j'$:

$$\Sigma_{ab} = \frac{1}{B^2} \cdot \sum_{j=1}^{N} \operatorname{cov}(T_j \, M_{i,j}, T_j \, M_{i,j}) g_{i,j}^{(a)} \, g_{i,j}^{(b)}$$

$$= \frac{1}{B^2} \cdot \sum_{j=1}^{N} \operatorname{var}(T_j \, M_{i,j}) g_{i,j}^{(a)} \, g_{i,j}^{(b)}$$

$$= \frac{1}{B^2} \cdot \sum_{j=1}^{N} \frac{B}{N} \cdot \left( 1 - \frac{B}{N} \right) g_{i,j}^{(a)} \, g_{i,j}^{(b)}$$

$$= \frac{1}{B \, N} \cdot \left( 1 - \frac{B}{N} \right) \sum_{j=1}^{N} g_{i,j}^{(a)} \, g_{i,j}^{(b)} \, .$$

In the above calculation, we use that $T_j \, M_{i,j}$ can be viewed as a Bernoulli random variable with parameter $\frac{N_t}{N} \cdot \frac{B}{N_t} = \frac{B}{N}$. Writing $\Sigma$ in the matrix form give us:

$$\Sigma = \frac{1}{B \, N} \cdot \left( 1 - \frac{B}{N} \right) \sum_{j=1}^{N} g_{i,j} \, g_{i,j}^{\top} \, . \tag{6}$$

Next, we compute $\Sigma^{(j,0)}$ and $\Sigma^{(j,1)}$ similarly for a fixed $j$. Condition on $T_j = 0$, $T_j \cdot M_{i,j}$ is always zero, so the variance is zero. Thus, we have:

$$\Sigma^{(j,0)} = \frac{1}{B \, N} \cdot \left( 1 - \frac{B}{N} \right) \sum_{j' \neq j} g_{i,j'} \, g_{i,j'}^{\top} \, . \tag{7}$$

Condition on $T_j$ being one, $T_j \cdot M_{i,j}$ is equal to $M_{i,j}$, a Bernoulli random variable with variance $(B/N_t) \cdot (1 - B/N_t)$. Thus, we have:

$$\Sigma^{(j,1)} = \frac{1}{BN_t} \cdot \left( 1 - \frac{B}{N_t} \right) \cdot g_{i,j} \, g_{i,j}^{\top} + \frac{1}{B \, N} \cdot \left( 1 - \frac{B}{N} \right) \sum_{j' \neq j} g_{i,j'} \, g_{i,j'}^{\top} \, . \tag{8}$$

## B  GNQ AS AN EXPLANATION FOR MODINV SUCCESS

We identified two classes of interest from the Database of Faces dataset: the class with the lowest GNQ (class 28) and the class with the highest GNQ (class 6). As shown in Fig. 9, the left image depicts poor reconstruction for the lowest GNQ class, while the right image demonstrates successful recovery for the highest GNQ class, showing increased vulnerability. Importantly, we noticed that high-GNQ examples are more likely to be *memorized*, which is validated as the reconstructed image (Fig. 9 b right) corresponds to the example (not class) with the highest GNQ in the entire dataset (Fig. 9 b left).

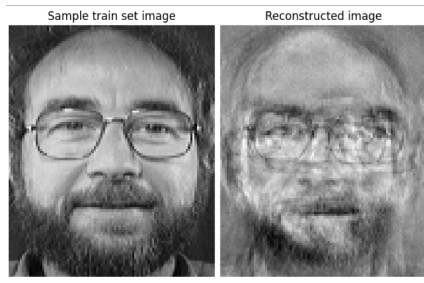
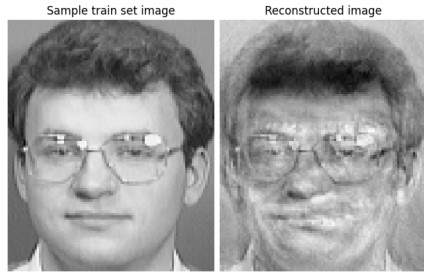

(a) Lowest-GNQ Class                    (b) Highest-GNQ Class

Figure 9: Reconstructed images for the lowest-GNQ class and the highest-GNQ class.

