# OpenReview forum: "Quantifying Information Disclosure During Gradient Descent Using Gradient Uniqueness"
_ICLR.cc/2026/Conference — Submitted to ICLR 2026_

### Official Review · Reviewer_98oZ · 2025-10-26

**Soundness:** 2
**Presentation:** 3
**Contribution:** 2
**Rating:** 2
**Confidence:** 5

**Summary:**

This paper introduces a new metric, "Gradient Uniqueness", designed to quantify the information disclosure of individual data points during mini-batch SGD. The metric is derived from an information-theoretic upper bound on the mutual information between a data point's membership and the final model parameters. The authors provide an intuitive explanation for GNQ and demonstrate empirically that it correlates with the success of Membership Inference Attacks and Model Inversion attacks. They also propose a simple defense: auditing the dataset with GNQ, removing the highest-risk points, and retraining. The results claim this defense achieves a better utility-privacy tradeoff than standard DP-SGD.

**Strengths:**

* Novel and Intuitive Metric: The core idea of "Gradient Uniqueness" is a novel contribution. The intuition that a point's privacy risk is related to how much its gradient "stands out" from the others in a batch is compelling. Section 4 provides an excellent, clear illustration of this concept.

* Practical Auditing Tool: The paper provides a concrete, computable metric for auditing privacy risk on a per-point basis. This is a practical and important problem, as developers often want to "test" their model for privacy vulnerabilities without paying the high utility cost of DP-SGD.

* Empirical Correlation: The experiments in Section 6.2 effectively demonstrate that the proposed GNQ score correlates well with the success of standard MIA and ModInv attacks across several datasets and models. This validates that the metric is capturing a real and meaningful signal of vulnerability.

**Weaknesses:**

* Vague and Weak Threat Model: The paper's primary weakness is its lack of a clear, explicit threat model. The setup (Line 132) assumes the adversary knows a public superset $\mathcal{D}$ and the private dataset $\mathcal{D}_t$ is sampled from it. This is a significant departure from standard security modeling, where an attacker is often assumed to know the full training set (for auditing) or the training algorithm must be robust to the addition/removal of any single point (for DP). This weak assumption makes it difficult to understand the true security provided and how it compares to DP-SGD. The paper would be significantly strengthened by a "Threat Model" section with a comparison table that contrasts its assumptions with those of other privacy works.

* Unprincipled Comparison to DP-SGD: The comparison in Table 1 is highly misleading and unprincipled due to to reasons. i) it compares the "privacy" (MIA AUC) of a heuristic defense (filtering) in a specific threat model against the provable, worst-case guarantee of DP-SGD, which operates in a much stronger threat model. A low MIA AUC is not equivalent to an $(\epsilon, \delta)$-DP guarantee. ii) The DP-SGD baseline results are poor (e.g., 41.83% on CIFAR10 and 6.83% on CIFAR100 for $\epsilon=2$). These results are far below standard published figures for DP-SGD and suggest the baselines were not properly tuned, see for example the results in [De et al. (2022)]. This makes the GNQ defense's utility look artificially high in comparison.

* Naive Defense and the "Onion Effect": The proposed defense (remove top-N points and retrain) is a simple heuristic, not a principled defense. This approach is known to suffer from the "onion effect" (see [Carlini et al. (2022)]). By removing the N most sensitive points, the (N+1)-th most sensitive point likely becomes vulnerable in the retrained model. The paper fails to investigate or even discuss this. A minimal check would be to re-run the GNQ audit on the filtered, retrained models.

* Vague Claims on Unlearning: The claim in Lines 057-060 that this technique can be used for "unlearning for open-source models" is confusing. If a model's weights are released "in the open," an adversary who has copied them cannot be forced to unlearn.

* Missing Related Work: The paper positions itself against DP-SGD but fails to engage with other relevant lines of work, such as other data-dependent information-theoretic privacy metrics (e.g., [Thudi et al. (2023)]) that provide exact measurements and can offer better utility-privacy tradeoffs than the naive DP-SGD baselines shown.

[De et al. (2022)]: https://arxiv.org/pdf/2204.13650
[Carlini et al. (2022)]: https://arxiv.org/abs/2206.10469
[Thudi et al. (2023)]: https://arxiv.org/abs/2307.00310

**Questions:**

* Threat Model: Could the authors please add a clear, standalone "Threat Model" section? As part of this, could you create a table comparing your assumptions (regarding dataset knowledge, attacker capabilities, and goals) to those in the standard DP, MIA, and gradient inversion literature?

* Dataset vs. Batch Privacy: Related to (1), can you clarify why the core assumption is that the dataset $\mathcal{D}_t$ is private (sampled from $\mathcal{D}$) rather than the more common assumption where the dataset $\mathcal{D}_t$ is known but the batch composition (the sampling for batches) is private?

* DP-SGD Baselines: Can the authors justify the direct comparison to DP-SGD, given the "apples to oranges" nature of an empirical metric vs. a provable guarantee? More importantly, could you please provide the hyperparameters and implementation details for the DP-SGD baselines, as the reported utility is far below published standards?

* Onion Effect: Regarding the proposed defense, did you re-calculate GNQ scores on the models that were retrained on filtered data? Does a new set of points become highly sensitive, as the "onion effect" would suggest? How many points must be removed before the model is considered "safe"?

* Open-Source Unlearning Claim: Can you clarify the statement in Lines 057-060? How, precisely, can this method "do unlearning for open-source models" after the weights are released and copied by an adversary?

---

### Official Review · Reviewer_VrtB · 2025-10-29

**Soundness:** 2
**Presentation:** 3
**Contribution:** 2
**Rating:** 2
**Confidence:** 4

**Summary:**

This paper introduces "gradient uniqueness" (GNQ), a new, principled metric for privacy auditing in models trained with standard mini-batch SGD. The authors provide a theoretical derivation that links GNQ to an information-theoretic upper bound on disclosure, based on a Gaussian gradient assumption. Empirically, GNQ is highly predictive of attack success. It correlates strongly with the success of both Membership Inference Attacks and Model Inversion attacks.  A simple defense—removing the top-p% of high-GNQ points and re-training—achieves MIA resistance comparable to DP-SGD

**Strengths:**

1. This paper focuses on an important question and proposes an interesting metric for privacy auditing.
2. The GNQ metric is well-motivated. The presentation is clear and easy to follow.
3. Experiments are clear and contain different models and datasets.

**Weaknesses:**

I think this paper's idea is good, but the experimental part needs to be improved.

1. Since you are proposing a new privacy auditing method, it will be more reasonable to compare it with some other auditing methods, such as Steinke, Thomas, Milad Nasr, and Matthew Jagielski. "Privacy auditing with one (1) training run." Advances in Neural Information Processing Systems 36 (2023): 49268-49280.
2. If you think you are proposing a new defense mechanism. Other MIA defense mechanisms, such as  SELENA, AdvReg, or HAMP, need to be the baseline.
3. For your Table 1, I wonder why the performance of DP-SGD is so bad even with high eps. Have you done a correct hyperparameter search for DP-SGD? Check the performance in this paper: De, Soham, et al. "Unlocking high-accuracy differentially private image classification through scale." arXiv preprint arXiv:2204.13650 (2022). You do not need to match their performance, but at least match their baseline performance.
4. How is the performance of your proposed method in terms of running time and memory usage?
5. The MIA you used is too old. Consider some new attacks like Ye, Jiayuan, et al. "Enhanced membership inference attacks against machine learning models." Proceedings of the 2022 ACM SIGSAC conference on computer and communications security. 2022.
Also, it is better to report TPR@low FPR as an evaluation metric for MIA.
6. I notice that your result is like a single run? Maybe an error bar for at least 3 runs is necessary to make the results more convincing.

**Questions:**

See the weakness.
I also wonder have you considered an alternative formulation as a privacy game? For example, a game between a "defender" (the trainer, whose payoff is model utility) and an "attacker" (whose payoff is the MIA success rate). Could such a game-theoretic model yield a direct theoretical bound on the optimal MIA success rate (e.g., the AUC), rather than a bound on information? This would provide a clearer theoretical link to the empirical results in Table 1 and Figure 5.

---

### Official Review · Reviewer_oVMr · 2025-10-31

**Soundness:** 2
**Presentation:** 3
**Contribution:** 2
**Rating:** 4
**Confidence:** 3

**Summary:**

This paper proposes a privacy disclosure metric for mini-batch gradient descent called gradient uniqueness (GNQ). GNQ does not make assumptions on model architecture, data type, or adversary strategy, and can be used to explain and predict the success rate of several privacy attacks. Based on this GNQ metric, the authors explored a defense that removes high-risk datapoints (according to GNQ) from training, and empirically showed that this GNQ-based defense can achieve comparable privacy as DP-SGD on CNN and transformer models while maintaining better model utility.

**Strengths:**

- The derivation of the proposed gradient uniqueness (GNQ) does not rely on a certain model architecture or data type.
- GNQ can be used to explain the performance of various attacks.
- Experiments considered various attacks (e.g., membership inference and model inversion attacks) and model architectures (e.g., MLP, CNN, and BERT).
- The paper is well-written.

**Weaknesses:**

- A major practical limitation of the proposed metric is its significant computational cost. As the authors acknowledge, the matrix S scales quadratically with the number of model parameters, making its computation and inversion “infeasible for a modern, large, model”. This cost is further compounded as S must be computed in a leave-one-out manner for each datapoint. While the authors suggest using smaller proxy models as an approximation, they provide no theoretical guarantees or empirical analysis on the error introduced by this heuristic.

- The theoretical analysis is grounded on the simplifying assumption that gradients follow a multivariate normal distribution, which is a strong assumption not guaranteed in practice. The practical implementation deviates even further from the theory by using heavy approximations, such as estimating only the diagonal of the S matrix. This effectively assumes all parameter gradients are uncorrelated. As such, GNQ is more accurately described as a principled heuristic in contrast to differential privacy, which provides a provable, worst-case bound on risk.

- The paper's analysis assumes an adversary who only observes the final released model parameters. While sufficient for one-time model releases, this is a weaker threat model than that often considered by DP-SGD, which can protect against adversaries who observe all intermediate gradient updates. This limits the applicability of GNQ as a defense in scenarios requiring multiple model releases, such as online learning, federated training, or unlearning [1].

- The authors propose a simple defense that filters training points with high GNQ scores. This approach, however, may introduce unintended risks. High-GNQ points may be outliers from the tail of the distribution that are critical for model generalizability, or they may represent a minority group, meaning their removal could harm model fairness. Furthermore, prior research [2] has identified a “privacy onion” effect, where removing the most vulnerable datapoints can expose a new layer of previously safe points. It remains unclear whether the GNQ-filtering defense would suffer from this same vulnerability.

- The experimental evaluation relies on average-case metrics, such as AUC ROC and test accuracy averaged over the entire dataset. This is a deviation from modern privacy literature [3, 4], which strongly advocates for measuring privacy risk in the worst case (e.g., for the most vulnerable examples or subgroups, which differs from the extreme worst case as in DP). Moreover, the experiments compare against a vanilla DP-SGD implementation. As some research [4] has suggested, such comparisons can be misleading, and a more robust evaluation would benchmark against state-of-the-art DP-SGD variants.

[1] Wu, Xiaoyu, et al. "Breaking the Gold Standard: Extracting Forgotten Data under Exact Unlearning in Large Language Models." arXiv preprint arXiv:2505.24379 (2025).
[2] Carlini, Nicholas, et al. "The privacy onion effect: Memorization is relative." Advances in Neural Information Processing Systems 35 (2022): 13263-13276.
[3] Carlini, Nicholas, et al. "Membership inference attacks from first principles." 2022 IEEE symposium on security and privacy (SP). IEEE, 2022.
[4] Aerni, Michael, et al. "Evaluations of machine learning privacy defenses are misleading." Proceedings of the 2024 on ACM SIGSAC Conference on Computer and Communications Security. 2024.

**Questions:**

Please refer to the weaknesses listed above.

---

### Official Review · Reviewer_Lu5A · 2025-10-31

**Soundness:** 3
**Presentation:** 3
**Contribution:** 2
**Rating:** 4
**Confidence:** 3

**Summary:**

The paper proposes a new privacy metric for gradient-based AI/ML model training, the gradient uniqueness (GNQ). For $g_{ij}$, gradient of data point $i$ at step $j$, it is simply given as
$$
GNQ_{ij} = g_{ij}^{\top} S^+ g_{ij},
$$
where $S^+$ is the Moore--Penrose pseudoinverse of the gradient covariance matrix at step $j$, i.e.,
$$
S = \sum_i g_{ij}^{\top} g_{ij}.
$$
The proposes strategies to for reducing the max GNQ by removing the data points with the biggest GNQ score. A theoretical connection between information theoretical data leakage and GNQ score is given. Experiments show that the method compares well against DP-SGD when measuring the actual privacy-utility tradeoffs (i.e., privacy leakage is measured via attacks, in this case via membership inference attacks).

**Strengths:**

- The measure is clear and intuitive, and the idea is of removing the most vulnerable data points via this measure is nice.

- Strong experimental results

**Weaknesses:**

- All the datasets in the experiments see to be balanced (equal number of datapoints per class) and the test accuracy over all test is measured. It would be interesting to see how these techniques affect different subgroups in case the dataset is unbalanced.

- The comparison to DP-SGD is not clear: what are the in and out samples for the membership inference tests. It is written “Privacy is measured using the success of an MIA attack on the resulting model, to identify the points that were left out of the training set.”, but I don't fully see what is included and excluded in case of GNQ.

- The hyperparameters of DP-SGD like clipping constant and learning rate etc. are not stated. I suspect they are not optimally chosen: DP-SGD with $\varepsilon=512$ ($\delta$ parameter is not stated) gives test accuracy 55\% for Cifar-10 while the baseline is over 80\%. Already, in the paper by [Abadi et al., 2016](https://arxiv.org/pdf/1607.00133), over 70\% test accuracy was obtained for $\varepsilon=4$. Since arguably the choice of the hyperparameters would affect the privacy-utility tradeoffs measured via membership inference attacks as well, I think it would be important to address this part.

- The theoretical result of Thm. A.1 and the discussion after it state that the upper bound on the information disclosure during the training grows monotonically as a function of the GNQ score. It is hard to make definite conclusions based on this result only (also experiments indicate that the relationship might not be monotonous, see Fig. 5). I think this theoretical statement would be much stronger if the monotonous relationship was actually shown in some sense, e.g., if there was a tight asymptotic relationship ($\Omega$-bounds in big-O notation).

**Questions:**

- The phenomenon that removing top 1 percent GNQ points sometimes drops the test accuracy by more than 3 percent (cifar 100) and sometimes increases the test accuracy (IMDB) is interesting. Do you have explanation for this?

- It is well known that the gradients of small subgroups are larger (more privacy leakage), and the model accuracy for those groups is smaller and vice versa (see e.g. [Yu et al., 2022](https://arxiv.org/pdf/2206.02617)). Considering that due to the pseudoinverse the singular values of the gradient covariance are inversed, one might expect that the scores would be the highest for small subgroups. Are the datapoints commonly removed from the smallest groups / groups with worst performance ?

- It is a little bit unclear how removing data points with the top 10 or 20 \% GNQ scores would affect the data remaining in the dataset. Would it increase their vulnerability?

- In all of the experiments of Fig. 5, the relationship between MI success rate and the GNQ score does not seem to be monotonous: there is kind of U-curve where for small GNQ scores the MI success rate is the same as for intermediate GNQ scores. Can you explain this? At least superficially, this seems to be in contradiction with the theoretical result that says that GNQ score increases monotonically as a function of the information disclosure as defined in the paper.

---

### Meta-Review · Area_Chair_ZWB5 · 2026-01-06

**Summary:**

Several concerns were raised in the initial reviews, and there were no rebuttals posted.

**Reviewer Scores:**

Unchanged

---

### Decision · Program_Chairs · 2026-01-26

Reject